# Post-Treatment of Tannic Acid for Thermally Stable PEDOT:PSS Film

**DOI:** 10.3390/polym14224908

**Published:** 2022-11-14

**Authors:** In-Seong Hwang, Ju-Yeong Lee, Jihyun Kim, Na-Young Pak, Jinhyun Kim, Dae-Won Chung

**Affiliations:** 1Department of Chemical and Materials Engineering, University of Suwon, Hwaseong 18323, Korea; 2EverChemTech Co., Ltd., 38, Cheongwonsandan 7-gil, Mado-myeon, Hwaseong 18543, Korea

**Keywords:** PEDOT:PSS, tannic acid, thermal stability, surface-treatment, electrical conductivity

## Abstract

As a poly (3,4-ethylenedioxythiophene) doped with poly (styrene sulfonate), PEDOT:PSS is well known for its conductive polymer in a field of organic electronics. PEDOT:PSS can be widely operated as electronics under low temperature conditions; however, the layer can be easily damaged by high temperature conditions, while in fabrication or in the operation of electronics. Therefore, enhancing the thermal stability of PEDOT:PSS can be a novel strategy for both fabrication and operating varieties. Herein, PEDOT:PSS is the surface-treated with tannic acid to increase the thermal stability. A large number of phenols in tannic acid not only provide UV absorption ability, but also thermal stability. Therefore, tannic-treated PEDOT:PSS film sustained 150 °C for 96 h because of its initial conductivity. Moreover, surface properties and its bonding nature was further examined to show that the tannic acid does not damage the electrical and film properties. The method can be widely used in the field of organic electronics, especially because of its high stability and the high performance of the devices.

## 1. Introduction

As a poly (3,4-ethylenedioxythiophene) doped with poly (styrene sulfonate), PEDOT:PSS is a heavily doped *p*-type conductive polymer that can disperse in water and provides a transparent film using a simple solution-coating process. Due to its unique combination of conductivity, transparency, flexibility, and high work function, PEDOT:PSS has been applied in various areas, such as antistatic layers, hole injection layers in organic light emitting diodes or solar cells, charge transfer layers on biomedical electrodes, and transparent electrodes for wearable devices [1,2,3,4]. Commercial products of transparent electrodes are usually made of doped metal oxides, such as indium or fluorine-doped tin oxide (ITO and FTO) [5,6,7,8,9,10,11,12,13,14]. Due to the scarcity of indium and the lack of flexibility of FTO and ITO, PEDOT:PSS is an attractive candidate as an alternative material to replace conventional electrodes [15,16,17,18,19,20,21,22,23,24,25,26,27,28,29].

However, PEDOT:PSS usually has two major drawbacks, namely low conductivity and thermal stability, as compared to the oxide-based electrode. Research to enhance the conductivity of PEDOT:PSS has been extensively carried out and the most classical and representative approach is by adding an organic compound, which is generally called a “secondary dopant”, to PEDOT:PSS. All these effects are well summarized in the earlier literature [30]. Approaches to improve water resistance have also been carried out and are mainly focused on modifying the acidic groups of PSS, the source of the hydrophilicity of PEDOT:PSS, by reaction with an epoxide containing a hydrophobic moiety, such as a epoxy silane coupling agent or epoxy compound with aromatic groups [31,32,33,34,35]. In particular, PEDOT:PSS reacted with silane coupling agents and was applied for a bio-electrode for electroencephalogram measurement and practically in-vivo tested for the brain activities [36,37,38,39].

However, there are relatively few studies to improve the thermal properties of PEDOT:PSS. In the general case of the PEDOT:PSS film, material degradation and resistance change occur within several hours. In general, it is known that the electrical properties of the PEDOT:PSS film are rather improved as the temperature increases from room temperature to 200 °C. This is explained as the decrease in moisture inside the film [39] or a change in morphology [40,41,42]. However, it is known that thermal decomposition occurs rapidly above 200 °C [14,39,43,44]. As the application of PEDOT:PSS becomes wider, the need for research into the long-term thermal stability of PEDOT:PSS is ever-growing.

In this study, we examined the thermal stability of the pristine PEDOT:PSS film and the effect of an additive, which is known to improve the electrical properties of the PEDOT:PSS film. These additives are generally called secondary dopants and most are liquid substances with high boiling points, such as DMSO and EG [30]. However, the innovative methods to enhance the electrical conductivity of PEDOT:PSS also include the use of water-soluble and naturally occurring materials like vitamins [45] or tannic acid (TA) [46]. Even though enormous studies were conducted to improve the conductivity of PEDOT:PSS, the stability study is not fully researched. Among various types of additives, we investigated the enhancement in thermal stability of PEDOT:PSS film by TA post-treatment, noting that the large numbers of phenol groups of TA, a type of polyphenol, provide not only UV absorption ability, but also thermal stability [47]. In this paper, we report the TA post-treated PEDOT:PSS film. The thermal stabilities of PEDOT:PSS film were largely improved after storage at 150 °C for 96 h. We analyzed surface morphologies of the films by AFM and the chemical composition of the films using the depth-profile method of XPS. Finally, we proposed the mechanism of the thermal stability of TA based on the change in chemical composition according to the depth of the film. This work demonstrates a method that induces amelioration in the thermal stability of PEDOT:PSS films without adversely affecting electrical properties.

## 2. Materials and Methods

### 2.1. Materials

PEDOT:PSS (CleviosTM P, Heraeus Holding GmbH, Hanau, Germany) was used without further purification. The solid content, viscosity, and weight ratio of PEDOT to PSS were 1.2%, less than 150 mPa·s, and 1:2.5, respectively. Tannic acid powder (Sigma-Aldrich (Burlington, MA, USA), dimethyl sulfoxide (DMSO, 99.0%, Sigma-Aldrich (Burlington, MA, USA), and silver paste (>75%, Sigma-Aldrich (Burlington, MA, USA) were purchased from Sigma-Aldrich.

### 2.2. Preparation of the Films

The effect of TA on the thermal stability of PEDOT:PSS were investigated in the following two ways: (1) The tannic acid powder was dissolved in PEDOT:PSS suspension. In this study, “1% addition” means the addition of 1 g of TA in 100 g of PEDOT:PSS dispersion. Glass slides with a dimension of 2 × 1 cm^2^ were cleaned with detergent, deionized water, acetone and methanol. The prepared suspension containing TA and PEDOT:PSS was spin-coated on the glass slides at 500 rpm for 10 s followed by 3000 rpm for 30 s. The films were dehydrated at 100 °C for 1 min. (2) PEDOT:PSS was spin-coated on the cleaned glass slides (2 × 1 cm^2^) at 500 rpm for 10 s followed by 3000 rpm for 30 s. After drying at 100 °C for 1 min, the films were immersed in a 1 wt% TA aqueous solution for 1 min and dried at 100 °C for 1 min.

To measure the linear resistance of the film, the silver paste was applied at 0.5 cm away from both ends of the films (i.e., the gap between the silver pastes is 1 cm). After drying for 1 h at 60 °C, the films were incubated in the convection oven at 150 °C in an air condition and the linear resistance (Ohm cm^−1^) between the two silver spots was measured every 24 h with a 2 point probe (289 True RMS Multimeter, Fluke). For further detailed analysis, the surface resistances of the films were also measured before and after incubation for 96 h.

### 2.3. Characterization

The morphologies of the PEDOT:PSS films were examined using atomic force microscopy (AFM; XE-100, Park Systems Co., Suwon, Korea) with a topographic image. XPS was recorded on a K-Alpha instrument (Thermo Scientific, Waltham, MA, USA), and the sputter condition was Ar2000 + 1.0 kV.

Resistance of the films was measured using a four-point probe (M4P-302, MSTECH, Hwaseong, Korea) setup with a source measurement unit (Keithley 2400, Tektronix, Beaverton, OR, USA). The values were converted into sheet resistance by taking correction factors into account. The thickness of films was measured by a surface profilometer (Alpha Step, P-7 of KLA-Tencor, KLA Corporation, Milpitas, CA, USA). To visually observe the thickness of the films, PEDOT:PSS or TA-treated PEDOT:PSS films were prepared on the ITO/glass by spin coating. A field-emission scanning electron microscope (Merlin Compact; Zeiss, Oberkochen, Germany) was utilized to observe the vertical structures of the films.

## 3. Results and Discussion

### 3.1. Thermal Stability

PEDOT:PSS is the conductive polymer that is widely used in organic electronics. However, the low thermal stability of PEDOT:PSS lead to poor device stability and fabrication limits. Tannic acid is a thermal stable molecule that contains multiple phenyl, hydroxyl and oxygen atoms, and it allows the additional interactions and protection with additives (Figure 1a). PEDOT:PSS is the polymer that PEDOT and PSS are interacting together, and the addition of tannic acid can interact with the PEDOT and PSS to increase the interaction strength against its well-known foe, heat (Figure 1b).

We examined the thermal stability of PEDOT:PSS by preparing a film in which TA was added a 1, 2, or 4 wt%, and a film obtained by dipping a PEDOT:PSS film in a 1, 2, or 4 wt% aqueous solution of TA. The linear resistance of the films was measured at 24 h intervals while stored at 150 °C, and the results are described in Figure 2.

Thermal treatment not only increases resistance drastically, but damages PEDOT:PSS film, which could be observed by bare eyes. However, TA treated films maintained relatively low resistance (even after thermal annealing). Before thermal annealing, linear resistance of 1% mixed and 1% solution dipped film denoted the value of 0.63 kΩ/cm and 0.71 kΩ/cm, respectively, which is similar to the pristine sample (0.65 kΩ/cm). In terms of 150 °C stored sample, linear resistance was investigated every 24 h and 1% or 2% of TA mixed sample resulted in the maintenance of low initial resistance, however, as storage time increases, the resistance increased undesirably. The TA solution dipped film showed increasing initial resistance with an increasing TA concentration but showed negligible discrepancy in the resistance induced by thermal annealing. To observe the quantitative electrical property, three samples from each condition were fabricated to scrutinize the surface resistance of before and after storage in specific conditions of 150 °C for 96 h.

As the concentration of TA increases and the thermal treatment had been conducted, incensement in standard deviation is obtained, which could be induced by the loss in uniformity of the film. A solution of 1% and 2% TA dipped film showed reliable data even after thermal treatment, which is proved by comparable results, and the ratio of recorded surface resistance and its initial value (R_s_/R_so_) of 1.24 or 1.20, while pristine PEDOT:PSS showed 1321.20, showing that thermal stability has been enhanced significantly. It is worth noting that the increment in initial resistance of 1% (0.92 kΩ/cm) was less than 20% of the pristine PEDOT:PSS result (0.77 kΩ/cm) demonstrating mitigated initial resistance. According to the aforementioned results, thermal stability in electrical properties of PEDOT:PSS was enhanced by TA, especially by dipping in the TA solution, rather than mixing, which turned out to be more effective in the surface treatment. Therefore, following the section aimed to investigate the mechanism of amelioration in thermal stability, mainly by a 1% TA solution dipped sample (Appendix A and Table 1).

### 3.2. Film Analysis

It is worth mentioning that there is a slight decrease in the conductivity of both TA treated samples. In particular, in the TA-dipped sample, a decrease in conductivity is a predictable outcome since the TA thin layer presents on the PEDOT:PSS layer, which results in a thickened layer. However, the TA in the thick layer did not affect the conductivity of the PEDOT:PSS film and largely improved the thermal stability, which could be the key strategy for TA modification. Briefly, as mentioned above, the TA-treated layer significantly prevents the segregation of PEDOT:PSS without damaging the electrical conductivity, especially in the PSS part, which leads to robustness in the chronic thermal stability issue of the PEDOT:PSS film (Figure 1, Figure 3 and Appendix A).

Secondly, we used AFM to investigate the differences in morphology of the pristine (PEDOT:PSS) and TA-PEDOT:PSS films before and after heat-treatment for the confined area of 5 µm × 5 µm, which is sufficiently large to provide a simultaneously reliable analysis.

As demonstrated in Figure 3a, the height image contains white and dark regions, which indicates that domains of different chemical components exist on the surface. It is customary that the bright region is read as PEDOT, and the dark region, as PSS [48,49,50]. The height image is almost the same as those of PEDOT:PSS reported earlier [36,38,44]. We also find that the root mean square roughness (R_rms_) of the pristine PEDOT:PSS film is 1.800 nm (measured over an area of 5 × 5 μm^2^), which is close to the reported values of 1.725 nm (5 × 5 μm^2^) [51]. However, the heat-treated film (Figure 3b) showed a distinct phase separation as if it was stained, and an obvious increase in R_rms_ from 1.800 to 4.390 nm. The significant non-uniformity and damage of the surface could also be visually confirmed.

Many research works have been focused on the improvement of the electrical or mechanical properties of PEDOT:PSS by the post-treatment. As summarized in the review paper [45], post-treatment of the films inevitably induced the change in the surface of the film. For example, treatment of PEDOT:PSS films with strong acids such as sulfuric acid induces significant phase separation and consequently improves the electrical properties of PEDOT:PSS films.

In Figure 3c, the roughness of PEDOT:PSS film after the surface-treatment of TA exhibited 3.703 nm, which is slightly higher than the value of PEDOT:PSS film without TA treatment. The roughness may change by TA, which may both penetrate and exist on the top of PEDOT:PSS film. However, TA post-treated PEDOT:PSS film after heat treatment resulted in negligible changes in surface roughness (Figure 3d), indicating that TA treatment protects the deformation of the PEDOT:PSS film under 150 °C for 96 h.

As a tentative conclusion, under the heat-treatment, a significant change in the surface morphology and the increase in surface roughness occurred for pristine PEDOT:PSS, however no meaningful change was observed for TA-dipped films.

### 3.3. XPS Studies by Depth-Profile

Since it is clear that thermal stability of the films mainly depends on the change in the surface of the film, we investigated the variation in the chemical composition using the depth-profile method of XPS using Argon sputtering.

As described in Figure 4a, C1s signals of pristine PEDOT:PSS showed main peak at 284.0 eV and a shoulder at 285.8 eV, which are attributed to C-C/C=C and C-O species, respectively [52,53]. After the first 100 s of sputtering, the peak remained in the same position and shape, however the peak barely exists after the second sputtering and disappears with the following cycle. It could be taken that the bottom layer of the film was reached after 200 s of sputtering. This pattern remained almost the same, even after the thermal treatment (Figure 4b). In the case of the TA treated PEDOT:PSS film, however, as shown in Figure 4c, besides the PEDOT:PSS intrinsic peak, the 288.0 eV and 285.4 eV peak were observed as well, which is representative of C=O and C-O in TA [54], respectively. The TA dominant thin layer from TA doping was formed on the top of the PEDOT:PSS surface and could disappear in one sputter cycle. Even after 1000 s of sputter, carbon peak remained the same, which is a good indication of thickened film (by TA treatment). Again, the pattern of heated TA-dipped PEDOT:PSS is the same with the unheated TA-dipped PEDOT:PSS (Figure 4d). The Si peak is additionally measured to address the thickness profile of the PEDOT:PSS film in Appendix A.

In general, insightful information on the chemical structure of PEDOT:PSS can be obtained from the S peak of XPS. The S 2p peak of PEDOT:PSS in the XPS spectra splits up into two regions. The peak from 162.0 to 167.0 eV is assigned to the PEDOT region and the peak from 164.0 to 172.0 eV is assigned to the PSS region. The area ratio (A(PEDOT)/A(PSS)) of these two peaks is the critical value to determine the electrical properties of PEDOT:PSS and the value is around 4.2 for the general PEDOT:PSS [51,55,56].

In Figure 5a, it is interesting that A(PEDOT)/A(PSS) is 0.21 for the top surface before etching, however the ratio dramatically increased to 5.62 after the 100 s of sputtering. This is considered to be that hydrophilic PSS segment is more present in the surface [57]. After 200 s of sputter, the S2p peak was hardly observed, which is consistent with C1s or the Si2p depth-profile result. Sputtering for 100 s partially brought a glass region and 200 s sputtering brought the matrix (glass) region, which could be reached by complete break-through of the film. By taking SEM and the alpha step together, about 20 nm was etched by one cycle of etching (100 s).

In the case of the thermal treated film (Figure 5b), the S2p peak from PEDOT became relatively undominant, which could be taken that thermal treatment causes degradation of PEDOT segment and lead to a deteriorated electrical property (also presented in Appendix A). The peak of the PEDOT never appears at the surface of the TA-dipped PEDOT:PSS film and this pattern also consists in the thermal treated sample (Figure 5c,d). In the case of the pristine sample, the thermal treatment affects not only the surface, but also the interior PEDOT domain, however by TA-doping, PEDOT remains in the sub-surface, which results in robustness in the electrical property. This indication is in good agreement with Figure 4 (C1s), namely that the TA dominant layer is present on the top of the surface and could be peeled by 100 s of sputtering, which is about 20 nm, estimated empirically by the abovementioned XPS and SEM data. In summary, in the case of pristine and thermal treated PEDOT:PSS, only 200 s of sputter brought the bottom glass surface.

In contrast, the TA doped film, which is consisted of the ~20 nm TA dominant layer and following well-ordered PEDOT and PSS layer (Figure 6), exhibited robustness even after 200 s of the sputtering cycle. The abovementioned data provides good indications that intrinsic electrical property has been maintained and improved thermal stability at the same time.

The results in Figure 1 and Figure 2 addressed that the TA-dipped PEDOT:PSS improves the thermal stability due to the involvement of TA in PEDOT:PSS. However, the results in SEM in Appendix A and XPS in Figure 4 and Figure 5 demonstrates that the TA may dominantly exist on the surface of PEDOT:PSS without harming the electrical properties (mechanical property and moisture stability can be the future work of the research). These observations elucidate that the TA-dipping method not only suppresses the segregation of PEDOT:PSS, but also hinders the conductivity reduction. Therefore, the TA-dipping can be utilized widely in the stable and highly performing devices.

## 4. Conclusions

PEDOT:PSS can be widely operated as electronics, such as solar cells, semiconductors and optoelectronics. However, thermal damage from the surrounding can cause the segregation of PEDOT and PSS, leading to the decrement in electrical conductivity and material stability of PEDOT:PSS. Therefore, methods that can enhance the thermal stability of PEDOT:PSS can be novel strategies for handling the PEDOT:PSS. By surface treatment with tannic acid, TA-treated PEDOT:PSS exhibits robust thermal stability. Therefore, TA-dipped PEDOT:PSS sustained 150 °C for 96 h for its initial conductivity. TA both inter-diffused and remained on a PEDOT:PSS, constructing the shielding interactions both inside and outside of PEDOT:PSS. The method can be widely appealed in the field of organic electronics, especially the high stability and the high performance of the devices.

## Figures and Tables

**Figure 1 polymers-14-04908-f001:**
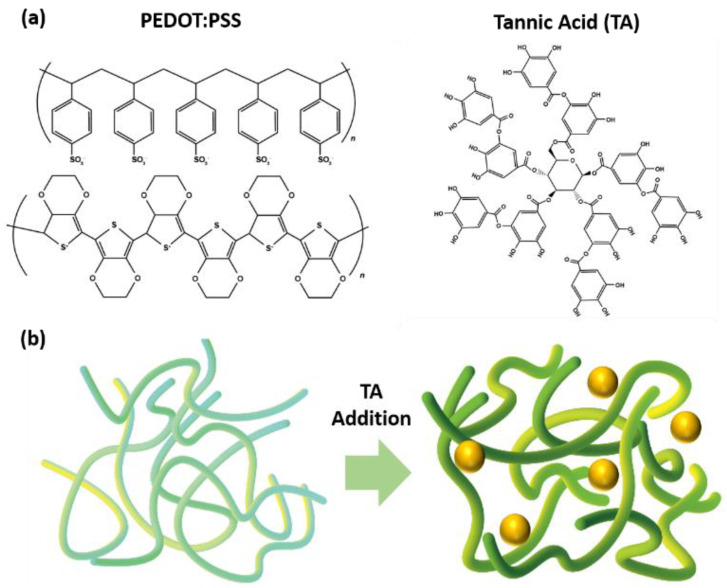
Chemical structures and effect of TA addition in PEDOT:PSS: (**a**) Structure of PEDOT:PSS and tannic acid, ad (**b**) changes of PEDOT:PSS via the addition of tannic acid.

**Figure 2 polymers-14-04908-f002:**
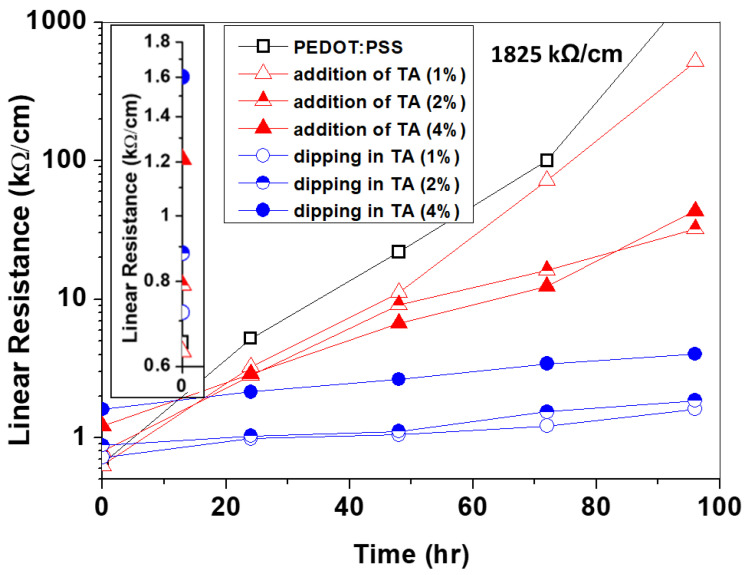
Linear resistance of PEDOT:PSS films as a function of incubation time at 150 °C. The insert illustrates the initial resistances of the films before incubation. 1825 kΩ/cm is the linear resistance of PEDOT:PSS after 96 h incubation.

**Figure 3 polymers-14-04908-f003:**
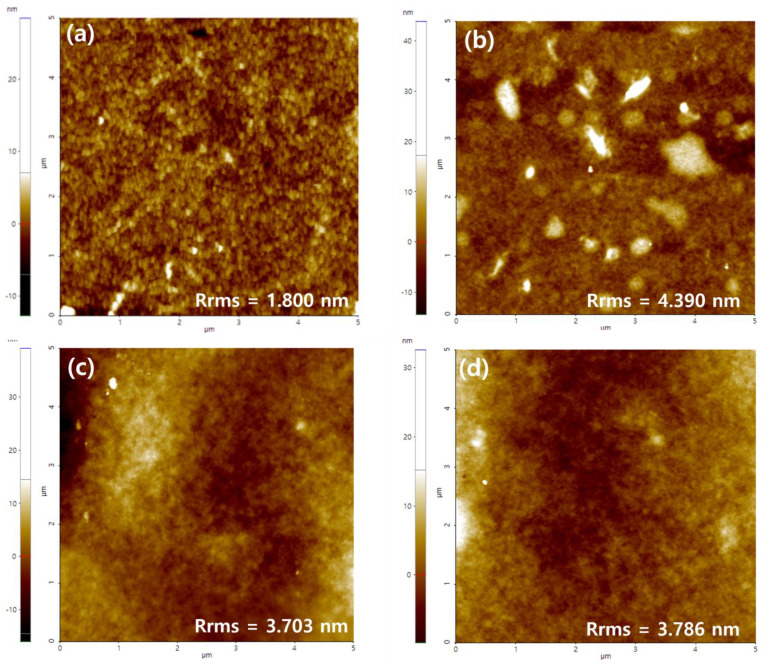
AFM surface images of (**a**) PEDOT:PSS, (**b**) PEDOT:PSS-HT, (**c**) TA-PEDOT:PSS, and (**d**) TA-PEDOT:PSS-HT films.

**Figure 4 polymers-14-04908-f004:**
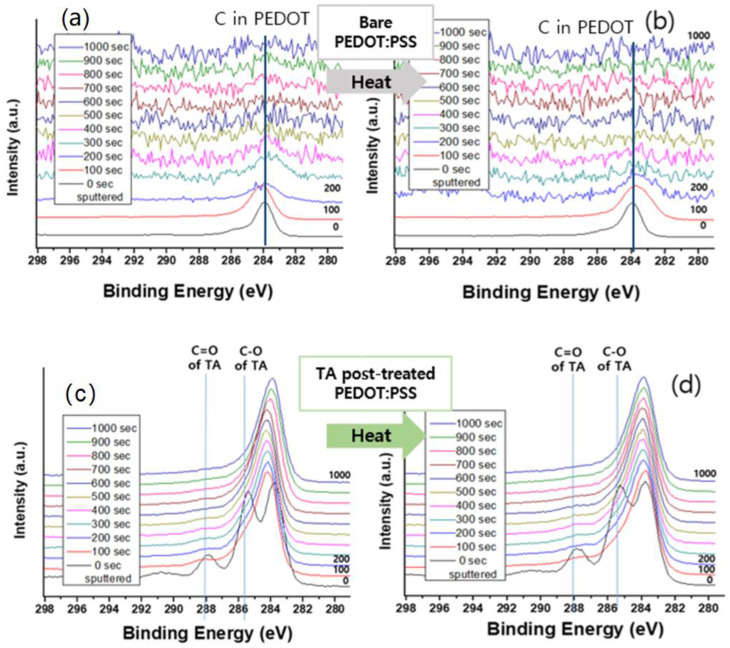
C1s XPS spectra of (**a**) PEDOT:PSS, (**b**) PEDOT:PSS-HT, (**c**) TA-dipped PEDOT:PSS, and (**d**) TA-dipped PEDOT:PSS-HT films. The legends indicate the time since the start of the depth profiling. Zero is equivalent to a standard surface scan.

**Figure 5 polymers-14-04908-f005:**
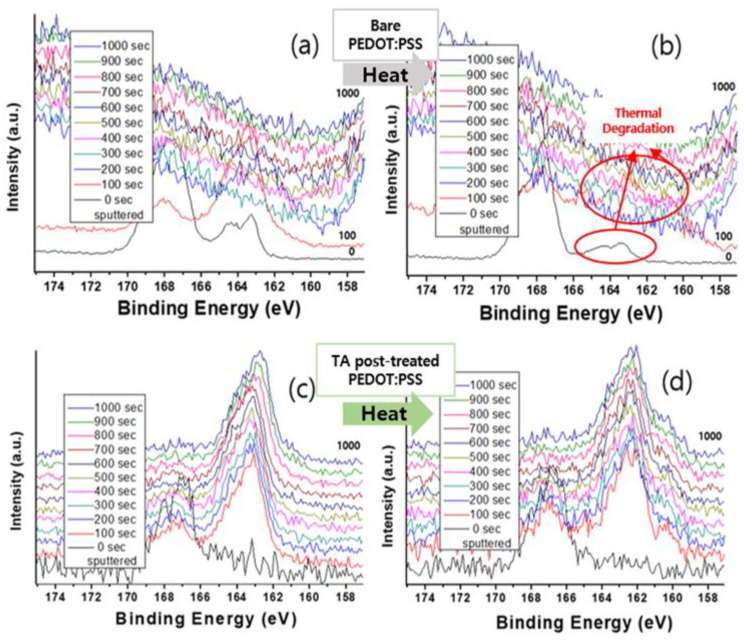
S2p XPS spectra of (**a**) PEDOT:PSS, (**b**) PEDOT:PSS-HT, (**c**) TA-dipped PEDOT:PSS, and (**d**) TA-dipped PEDOT:PSS-HT films. The legends indicate the time since the start of the depth profiling. Zero is equivalent to a standard surface scan.

**Figure 6 polymers-14-04908-f006:**
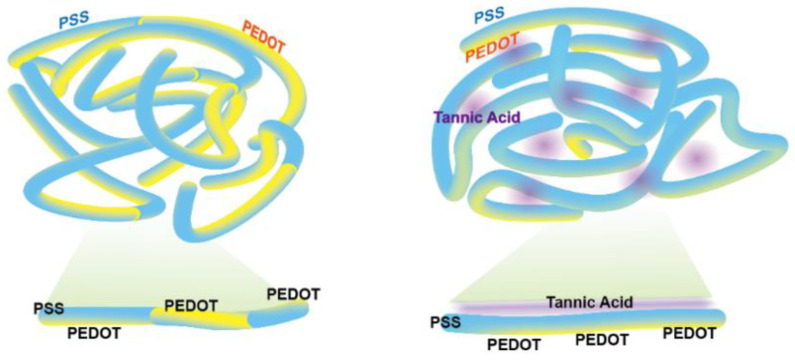
Schematics for chemical and film morphology of TA-treated PEDOT:PSS: (**left**) Structure of bare PEDOT:PSS, and (**right**) changes of PEDOT:PSS via treatment of the TA.

**Table 1 polymers-14-04908-t001:** Surface resistance and thickness of PEDOT:PSS films treated with TA before and after incubation at 150 °C for 96 h.

	Concentration of TA	Surface Resistance (kΩ/sq)			Film Thickness (nm)			
					By α-Step		By SEM	
		Before Incubation * (Rs_o_)	After Incubation * (Rs)	Rs/Rs_o_	Before Incubation *	After Incubation *	Before Incubation	After Incubation
Pristine	-	0.77 ± 0.05	1020.33 ± 230.38	1321.10	38 ± 7	36 ± 9	35.7	37.7
Addition of TA	1%	0.76 ± 0.05	880.33 ± 221.38	1154.79	79 ± 18	75 ± 17	67.0	71.4
	2%	1.04 ± 0.19	10.49 ± 3.34	10.12	206 ± 26	209 ± 38	-	-
	4%	1.84 ± 0.77	11.89 ± 1.89	6.46	397 ± 79	372 ± 84	-	-
Dipping in TA solution	1%	0.92 ± 0.12	1.15 ± 0.16	1.24	302 ± 26	261 ± 41	357.3	332.6
	2%	1.05 ± 0.17	1.26 ± 0.26	1.20	428 ± 62	406 ± 80	-	-
	4%	4.20 ± 1.59	5.19 ± 1.43	1.23	448 ± 98	423 ± 90	-	-

*: The average and standard deviation values were calculated from three different samples independently prepared.

## Data Availability

The data presented in this study are available on request to the corresponding author.

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
