# Peer review of "Post-Treatment of Tannic Acid for Thermally Stable PEDOT:PSS Film"

_polymers, 2022, doi:10.3390/polym14224908_

Round 1

Reviewer 1 Report (New Reviewer)

Dear authors,

Here are my comments:

1. Which are the novelty and added value of the paper with respect to existent literature?

2. FTIR/RAMAN analysis is important.

3. Thermal analysis could add value: TGA, DSC. It could be nice to check the influence of TA on PEDOT:PSS films on TGA and DSC.

4. SEM morphology should be added in the paper together with AFM.

5. What about the mechanical properties of the materials (PEDOT:PSS vs TA-PEDOT:PSS).

6. Did you check the free surface energies of the films (contact angle measurements)?

Author Response

Dear Reviewer:

Please find the revised manuscript submitted to the Polymers entitled, “Post-treatment of Tannic Acid for Thermally Stable PEDOT:PSS Film”

Regarding the decision on the manuscript (Research Article, No. polymers-2007074), we hereby include detailed responses to the reviewers’ comments.  We were able to improve our manuscript based on the reviewers’ helpful and constructive comments.  Changes for the revision are highlighted with yellow color in the revised manuscript.  To address the reviewers’ comments, the manuscript is revised as follows:

[Reviewer 1]

1. Which are the novelty and added value of the paper with respect to existent literature?

Thank you for the reviewer’s kind comment.  As discussed in introduction, additive studies in PEDOT:PSS is quietly searched to improve the electrical conductivity [30,45,46].  As far as we know, it is first time that relating the thermal stability and conductivity of PEDOT: PSS films with surface-treatment.  In order to commercialize, stability study is crucial and our work have a potential to widen the application of polymer electronics.  To clarify the novelty of our work, additional explanation is added on the last paragraph in introduction with a yellow mark.

[30] Ouyang, J. “Secondary doping” methods to significantly enhance the conductivity of PEDOT:PSS for its application as transparent electrode of optoelectronic devices. Displays. 2013, 34, 423-436.

[45] Huseynova, G.; Kim, Y.H.; Lee, J.-H.; Lee, J.H. Rising advancements in the application of PEDOT:PSS as a prosperous transparent and flexible electrode material for solution-processed organic electronics. J. Infmater.. Disp. 2020, 21, 71-91.

[46] Yi, Z.; Zhao, Y.; Li, P.; Ho, K.; Blozowski, N.; Walker, G.; Jaffer, S.; Tjong, J.; Sain, M.; Lu, Z. The effect of tannic acids on the electrical conductivity of PEDOT: PSS Films. App. Sur. Sci. 2018, 448, 583-588.

2. FTIR/RAMAN analysis is important.

As reviewer commented, RAMAN analysis is additionally measured (Figure S3).  Figure S3a revealed that degradation of PEDOT:PSS after heated.  C-C stretching peak I PEDOT:PSS is largely suppressed after heated, indicating the degradation is occurring.  On the other hand, TA post-treated PEDOT:PSS resulted unchanged peak when the peak is compared before and after thermal induction.  However, data from TA-treated PEDOT:PSS did not showed PEDOT:PSS peak, and it may influenced by the beam depth of RAMAN spectroscopy.  Because TA is surface treated, surface may include TA dominantly.  We believe that the data in Figure 5 afford more accuracy for the thermal degradation behavior of PEDOT:PSS films.  XPS is conducted with depth profiling analysis, we can detect the chemical species from surface to deeper level of film.  Thanks to reviewer for the constructive comment.

3. Thermal analysis could add value: TGA, DSC. It could be nice to check the influence of TA on PEDOT:PSS films on TGA and DSC.

As reviewer kindly commented, TGA and DSC could give the additional data for thermal stability.  However, the our research is focused on thermal stability of PEDOT:PSS film after post-treating the TA.  Therefore, TGA and DSC cannot be measured in a film system.  If our research is done by mixing of PEDOT:PSS and TA, TGA could be conducted.  However, TA is post-treated on a surface of film, which should be measured via film analysis.  Many appreciates for reviewer’s kind comment.

4. SEM morphology should be added in the paper together with AFM.           

Thank you for reviewer’s comment.  As reviewer discussed, SEM could give an additional morphology.  However, AFM is more practical to understand the roughness of the film and the additional SEM image is presented in Figure S1.  As shown in Figure S1, overall PEDOT:PSS layer is in a dark area, indicating that the PEDOT:PSS is hard to observe using SEM.

5. What about the mechanical properties of the materials (PEDOT:PSS vs TA-PEDOT:PSS).           

Thank you for reviewer’s consideration.  As discussed in the last paragraph on result and discussion session, TA is not chemically bonded with PEDOT:PSS (Figures 4 and 5).  Therefore, it is surface and mixed as a composite form.  Therefore, there will be small changes in mechanical property but it would be worth it as a future research.  Additional comment is added on a last paragraph (result and discussion session) on page 10.

6. Did you check the free surface energies of the films (contact angle measurements)?           

Both tannic acid and PEDOT:PSS is well known for hydrophilic material.  Therefore, there will be no difference between PEDOT:PSS and TA treated PEDOT:PSS.  Moisture stability can be a new topic for another research.  Additional comment is added on a last paragraph (result and discussion session) on page 10.  Thank you for reviewer’s worries and consideration.

Authors’ Additional Modification

To further improve the quality of the manuscript, English and format are checked once more throughout the manuscript, and some minor necessary corrections are also made.Sincerely yours,

Jinhyun Kim

Assistant Professor

Reviewer 2 Report (New Reviewer)

     The manuscript: “Post-treatment of Tannic Acid for Thermally Stable PEDOT:PSS Film” by In-Seong Hwang et al., reports the results of preparation and investigation of the post-treated tannic acid (TA) on PEDOT:PSS film which is used in electronics under low temperature conditions. It was shown that the thermal stability of PEDOT:PSS  film was largely improved after storage at 150°C for 96 hours. The authors analyzed surface morphologies of the films by AFM and the chemical composition of the films by depth-profile method of XPS. It was proposed that the mechanism of the thermal stability of TA based on the change in chemical composition according to the depth of the film. The results demonstrate a method to improve the thermal stability of PEDOT:PSS films without adversely affecting electrical properties.

    The results presented by the authors may be of interest for research in the field of conducting polymers and polymer electronics. In my opinion, the topic and materials of this study is up-to-date. At the same time, the following main questions remain.

1.     I recommend emphasizing the novelty of this study more clearly in the introduction and conclusion to show the contribution of the authors in this field obtained in this study.

  1. There are several questions regarding the conditions of the experiment. The authors noted that “The thermal stabilities of PEDOT:PSS film were largely improved after storage at 150°C for 96 hours”. The question arises: under what natural conditions were these measurements carried out: were they carried out in air or in an inert atmosphere?
  2. In fig. 2 shows the linear resistance of PEDOT:PSS films as a function of incubation time at 150°C. I think that the dependence of DC conductivity on incubation time will better show the difference between different samples.
  3. I also propose to present current-voltage (I-Vs) of such samples. It is important to ensure that all resistance values have been calculated from the linear mode of these IVs.
  4. Page 7, line 9 on top. I don't understand the following sentence: " However, in this study, TA post-treatment on PEDOT:PSS films may affect morphology of the film and also induced the increase in the surface roughness (Rrms) to from 3.703 (Figure 3d)". .

What is this “…from 3.703…”?

  1.  Finally, I strongly recommend improving the style of this manuscript, especially in the Introduction. English also needs to be improved.

In conclusion, I believe that the topic of this manuscript may be in line with that of the Polymers. At the same time, the manuscript needs some improvement. Authors should revise the manuscript strictly in accordance with the guidelines mentioned above. A revised manuscript can be resubmitted for publication in Polymers.

Author Response

Dear Reviewer:

Please find the revised manuscript submitted to the Polymers entitled, “Post-treatment of Tannic Acid for Thermally Stable PEDOT:PSS Film”

Regarding the decision on the manuscript (Research Article, No. polymers-2007074), we hereby include detailed responses to the reviewers’ comments.  We were able to improve our manuscript based on the reviewers’ helpful and constructive comments.  Changes for the revision are highlighted with yellow color in the revised manuscript.  To address the reviewers’ comments, the manuscript is revised as follows:

[Reviewer 2]

1. I recommend emphasizing the novelty of this study more clearly in the introduction and conclusion to show the contribution of the authors in this field obtained in this study.

Many thanks for the reviewer’s kind comment to improve the quality of our work.  There are many studies to improve the electrical conductivity of PEDOT:PSS film [30, 45, 46].  However, understanding thermal degradation and methods for improvement is yet not fully discovered.  Our work has much potential to understand the thermal behavior of PEDOT:PSS film.  To clarify the novelty of our work, additional explanation is added on the last paragraph in introduction with a yellow mark.

[30] Ouyang, J. “Secondary doping” methods to significantly enhance the conductivity of PEDOT:PSS for its application as transparent electrode of optoelectronic devices. Displays. 2013, 34, 423-436.

[45] Huseynova, G.; Kim, Y.H.; Lee, J.-H.; Lee, J.H. Rising advancements in the application of PEDOT:PSS as a prosperous transparent and flexible electrode material for solution-processed organic electronics. J. Infmater.. Disp. 2020, 21, 71-91.

[46] Yi, Z.; Zhao, Y.; Li, P.; Ho, K.; Blozowski, N.; Walker, G.; Jaffer, S.; Tjong, J.; Sain, M.; Lu, Z. The effect of tannic acids on the electrical conductivity of PEDOT: PSS Films. App. Sur. Sci. 2018, 448, 583-588.

2. There are several questions regarding the conditions of the experiment. The authors noted that “The thermal stabilities of PEDOT:PSS film were largely improved after storage at 150°C for 96 hours”.  The question arises: under what natural conditions were these measurements carried out: were they carried out in air or in an inert atmosphere?

The condition for the thermal test (150°C for 96 hours) was done under air condition.  To clarify the experiment, the detail of heat test is additionally discussed on the second paragraph in an experimental section with yellow marked.  Many appreciates for the improving the quality of our work.

3. In fig. 2 shows the linear resistance of PEDOT:PSS films as a function of incubation time at 150°C. I think that the dependence of DC conductivity on incubation time will better show the difference between different samples. I also propose to present current-voltage (I-Vs) of such samples. It is important to ensure that all resistance values have been calculated from the linear mode of these IVs.

Many appreciate for the reviewer’s kind comment.  I-V characteristic surely guarantee the charge transport behavior of the organic electronics.  However, application as electronics is remained as the future work due to the difficulty of the device fabrication and optimization.

4.Page 7, line 9 on top. I don't understand the following sentence: " However, in this study, TA post-treatment on PEDOT:PSS films may affect morphology of the film and also induced the increase in the surface roughness (Rrms) to from 3.703 (Figure 3d)". .

As reviewer commented, the sentence had some error to deliver the message we wanted to present.  We have rewritten the sentence to clarity our statement on the first paragraph in page 7 with yellow marked.  Due to the quality of surface is highly important in both device and film stability, treatment of TA on PEDOT:PSS film is highly important to observe non-changing morphology.   Many appreciate for reviewer for correcting the error.

5. Finally, I strongly recommend improving the style of this manuscript, especially in the Introduction. English also needs to be improved.

Thank you for reviewer’s critical comment.  Introduction was rewritten and deleted the some irrelevant sentences to clearly address the thermal stable effect of TA post-treatment on PEDOT:PSS film.  Changes are made on introduction with the yellow marked.  Many appreciates for reviewer’s comment.

Authors’ Additional Modification

To further improve the quality of the manuscript, English and format are checked once more throughout the manuscript, and some minor necessary corrections are also made.

Sincerely yours,

Jinhyun Kim

Assistant Professor

Round 2

Reviewer 1 Report (New Reviewer)

Dear authors,

The paper has been improved and it could be accepted in Polymers journal. I would recommend a supplimentary English check-up and a stronger focus on the novelty and added value.

This manuscript is a resubmission of an earlier submission. The following is a list of the peer review reports and author responses from that submission.

Round 1

Reviewer 1 Report

The article title title, Thermally Stable PEDOT:PSS via Tannic Acid Utilization as an Efficient Polymer Electrode, studied the thermal stability of PEDOT:PSS as treated with tannic acid. The article does not provide a clear explanation at the molecular level of the thermal stability improvement of the PEDOT:PSS films. The result are highly difficult to follow with many phrases not supported by the analysis done on the samples. The authors do not provide other thermal analysis (TGA) that can be clarify why tannic acid improved the thermals stability of PEDOT. The name of the samples change on every graph making more difficult to follow the results. There is not discussion comparing the result with current literature on the subject. The article present low soundness to the scientific audience and does not present any novelty to the conductive polymer field. I regret to say that I do not suggest the publication of this article.  

Reviewer 2 Report

The authors addressed some interesting experimental results in this manuscript, However, the overall quality of the writing does not meet the expectation, and additional validation must be considered; therefore, this version should not be published in MDPI Polymers.